# Resource-constrained Neural Architecture Search on Language Models: A Case Study

**Andreas Paraskeva** [1]  **João Pedro Reis** [2]  **Suzan Verberne** [1]  **Jan N. van Rijn** [1]

## Abstract

Transformer-based language models have achieved milestones in natural language processing, but they come with challenges, mainly due to their computational footprint. Applying automated machine learning to these models can democratize their use and foster further research and development. We present a case study using neural architecture search (NAS) to optimize DistilBERT in a resource-constrained environment with a $4\,000$ GPU-hour budget. We employ an evolutionary algorithm that uses a two-level hierarchical search space and a segmented pipeline for component enhancement. While in order to obtain state-of-the-art results more compute budget is required, our results show efficient exploration, and a strong correlation between pre-training and downstream performance. This suggests a potential applicability of using pre-training validation as a cutoff criterion during model training. Finally, our learning curves analysis emphasizes the potential for efficient resource allocation through the adoption of an epoch-level stopping strategy, thus directing resources towards more promising candidate models. Future work should focus on scaling these insights to larger language models and more diverse tasks.

## 1. Introduction

Recent advancements in the field of **L**arge **L**anguage **M**odels (LLMs) (Zhao et al., 2023) have caught the attention of many researchers and users. The transformer

architecture (Vaswani et al., 2017) and subsequent models such as **B**idirectional **E**ncoder **R**epresentations from **T**ransformers (BERT) (Devlin et al., 2019), **G**enerative **P**re-trained **T**ransformer (GPT) (Radford & Narasimhan, 2018), BLOOM (Scao et al., 2022) and LLaMa (Touvron et al., 2023) introduced rapid breakthoughs in the field of **N**atural **L**anguage **P**rocessing (NLP). Platforms such as Hugging-Face empower developers, researchers, and organizations by providing pre-trained models, fostering innovation and community-driven research.

The key idea of **Auto**mated **M**achine **L**earning (AutoML) (Hutter et al., 2019) is to is to optimise several aspects of the training process such as the neural architecture. AutoML plays a supporting role in automating the identification of an appropriate machine learning pipeline for a given task. The underlying aim is the democratization of machine learning, enabling researchers to apply their expert knowledge to higher-value tasks while automating routine processes such as hyperparameter optimization and the search for a better neural architecture. Designing effective neural network architectures and tuning structural parameters is a challenging and widely researched area. These challenges have led to the inception of the field of **N**eural **A**rchitecture **S**earch (NAS) (White et al., 2023), where the specific task of designing an architecture is automated.

Challenges more specific towards language models also arise. The LLM training procedure consists potentially of two phases, (i) a pre-training phase on a large corpus of (often unlabelled) text, which is typically computationally expensive, and (ii) a fine-tuning phase on the task of interest, which is computationally much cheaper. The limited intuition towards their effective design is an obstacle that also restricts the explored architectures since humans are less likely to investigate more complex architectures. Indeed, while NAS has the potential to identify improved models, NAS on the pre-training of language models has been identified as one of the most complex challenges for deploying AutoML on LLMs (Tornede et al., 2023).

In our research, we investigate transformer-based language models, typically characterized by a large number of trainable parameters. The objective is to explore the macro-architecture of these models and derive an effective search

---

[1]Leiden Institute of Advanced Computer Science (LIACS), Leiden University, Leiden, Netherlands [2]Departamento de Engenharia Eletrotécnica e de Computadores, University of Porto, Porto, Portugal. Correspondence to: Andreas Paraskeva <a.paraskeva@liacs.leidenuniv.nl>.

Accepted to the Workshop on Advancing Neural Network Training at International Conference on Machine Learning (WANT@ICML 2024).

space for potential models to explore. Additionally, we want to adopt a search method that utilises a multi-objective optimization approach. We hope to primarily optimize the performance of the model on the downstream task of question answering, while secondarily minimizing the explored candidates' model sizes. Our overall aim is to investigate the application of AutoML to LLMs, acknowledge arising challenges (Tornede et al., 2023), and identify promising opportunities for making this procedure more efficient.

In this paper, we report on a case study that brings NAS and language models together in a resource-constrained environment. We apply our proposed NAS methodology to the DistilBERT model[1] (Sanh et al., 2019), which was pre-trained on the task of **M**asked **L**anguage **M**odeling (MLM). While larger models exist, due to the computational cost of pre-training such models, we choose this relatively smaller language model of approximately 66 million parameters. As such, we will refer to DistilBERT as a language model, rather than an LLM. The search space includes two searchable levels by adding macro-level architecture hyperparameters to a cell-based search; derived from submodules of the transformer encoder (i.e. Feed-forward Neural Network and Multi-head self-attention). The search method is based on a genetic algorithm (Bäck, 1996), specifically an adapted version of **N**on-dominated **S**orting **G**enetic **A**lgoritm II (NSGA-II) (Deb et al., 2002) to optimize generated models on the two previously mentioned objectives (performance and model size). We perform pre-training and finetuning on explored model variants, and use the downstream task performance metric as one of the objective scores.

Based on the expensive life-cycle of transformer-based language models, in combination with the resource-constrained environments that are typically available to researchers, it is important to use available resources effectively and efficiently. In such a resource-constrained environment, we utilised a small compute grant out of which of $\sim$4 000 GPU-hours have been spent on our pipeline, while an extra portion GPU-hours was dedicated to development purposes and preliminary analysis. The used GPUs were *NVIDIA A100*s with 40GB VRAM. This execution would otherwise amount to approximately €15 000 assuming use of a popular cloud platform.[2] Through our experiments, we derive insights into typical resource needs and identify potential areas for targeted focus and cost-effective resource allocation.

While in order to obtain state-of-the-art results more compute budget is required, our results show efficient explo-

ration of the search space. Firstly, we observe a strong correlation between pre-training and performance on the downstream task, suggesting a potential applicability of using multi-fidelity approaches as a cutoff criterion during the pre-training. Secondly, a learning curves analysis emphasizes the potential for efficient resource allocation through the adoption of an epoch-level stopping strategy, thus directing resources toward more promising candidate models. The above observations indicate potential in applying multi-fidelity approaches (such as learning curve analysis (Mohr & van Rijn, 2022) or bandit-based methods (Li et al., 2017)) to early discard non-promising candidate networks, which has the potential to speed up the search process tremendously.

We see this work as one of the first stepping stones towards performing efficient NAS on the pre-training of language models for specific tasks.

## 2. Related Work

We structure the related work in sections based on transformer-based language models, NAS, and NAS for large language models.

### 2.1. Transformer-based Language Models

Pre-trained language models, particularly transformer-based architectures (Vaswani et al., 2017), have revolutionized the field of natural language processing. Amongst the most influential LLMs, the GPT (Radford & Narasimhan, 2018) has showcased the potential of generative pre-training. BERT (Devlin et al., 2019) uses bidirectional transformers, through stacking the encoder of the transformer and a masked language model objective. BART (Lewis et al., 2020) and PaLM (Chowdhery et al., 2023) used in context learning, whilst Megatron-Turing NLG (Smith et al., 2022) used reinforcement learning. There are various possible learning paradigms for the pre-training phrase, but for this work, we will focus on MLM which was also used for the pre-training phase of DistilBERT from HuggingFace.

### 2.2. Neural Architecture Search (NAS)

A NAS methodology heavily depends on the expressiveness of the search space, the effectiveness of the search method that traverses it, and the appropriate usage of performance estimation strategy (White et al., 2023). Early adopters of NAS were based on reinforcement learning (Zoph & Le, 2017) which possessed clear downsides in terms of computational cost. Notably, **E**fficient **N**eural **A**rchitecture **S**earch (ENAS) (Pham et al., 2018) is an efficient reinforcement learning-based NAS algorithm that leverages a supernet to optimize the search space. By training a supernet that contains all possible subnetworks, the algorithm can efficiently identify the best architecture for a given

---

[1]This is the distilled version of Bert Base uncased provided by https://huggingface.co/distilbert-base-uncased, which is trained on English language using a masked language modeling objective.

[2] Assuming usage of a machine with two *A100* GPUs, the cost is calculated according to the respective pricing in the *Google Cloud* and *Amazon Web Services (AWS)*.

task. Gradient-based methods, like **D**ifferentiable **A**rchitecture **S**earch (DARTS) (Liu et al., 2019), utilised continuous relaxation of the search space, making it differentiable and thus enabling the use of gradient descent. These supernet-based approaches speed up the model search at the cost of additional model size during training, which can be restrictive in situations where memory is a limiting factor (such as in the case of training an LLM). **E**volutionary **A**lgorithm (EA) based methods were proposed to exploit evaluated candidate models for effective population maintenance and evolution (Jian et al., 2023; Deb et al., 2002). An interesting approach was NSGA-Net (Lu et al., 2019), which employed a multi-objective genetic algorithm to efficiently balance accuracy and computational cost in NAS for **C**onvolution **N**eural **N**etworks (CNNs). This does not directly translate to transformer-based models but strengthens the potential applicability of utilised techniques.

## 2.3. NAS for LLMs

NAS has been heavily researched in the area of CNNs and **R**ecurrent **N**eural **N**etworks (RNNs), with a focus on the computer vision field. Although research on NAS for transformers has been limited, it is gradually expanding, especially into NLP-related tasks. For example, Primer (So et al., 2021) is a NAS algorithm based on EA that searches for a decoder-only auto-regressive language model. The search space definition consists of TensorFlow applications (primitives) and evolution is applied to search for candidate models. *AutoBERT-Zero* (Gao et al., 2022) proposes a search space containing primitive operations with the NAS method aimed to develop a novel attention structure. *NAS-BERT* (Xu et al., 2021) employs neural architecture search to compress BERT models, achieving adaptive model sizes and task-agnostic applicability. The main drawback would be the high computational cost associated with training a large supernet. *AutoTinyBERT* (Yin et al., 2021) focused on the automatic optimization of hyperparameters defining the architecture of BERT, which was tailored towards resource-constrained environments. Following a macro-search space it contained limited expressiveness but depicted very promising results. Inspired by the *AutoTinyBERT*, we developed a more expressive hierarchical search space, that defines primitive components of the transformer encoder backbone, and added macro-level architecture hyperparameters in order to mutate the current architecture in a modular approach. Additionally, we retain weights associated with unaffected layers post-mutation (Real et al., 2017) and investigate the effectiveness of epoch-level multi-fidelity in our NAS method.

## 3. Methodology

In this case study, we propose a definition for the NAS search space, accompanied by a search method responsible

for the exploration of models within it.

### 3.1. Search Space

The architecture of a transformer encoder in the DistilBERT model has four components: *Multi-head self-attention*, *Layer Normalization*, *Feed-forward Neural Network layer (FNN)* and the *output layer*. The multi-head self-attention allows the model to focus on different parts of the input sequence simultaneously, capturing relationships between tokens. The FNN introduces non-linearity, allowing the model to capture complex patterns and high-level features. Arguably the multi-head self-attention and FNN are the most crucial parts of the transformer encoder. We introduce enhanced searchability to a cell-based approach by incorporating macro-level architecture hyperparameters; thereby incorporating two levels of searchability in a hierarchical search space.

Firstly, the multi-head self-attention mechanism involves the specification of the number of attention heads, each focusing on distinct aspects or patterns in the data. This parallelization enables a richer understanding of complex data patterns, albeit with increased computational and memory costs. Secondly, the FNN incorporates several hyperparameters. The choice of activation function, namely ReLU (Agarap, 2018) or GELU (Hendrycks & Gimpel, 2016), impacts the non-linear transformations between linear layers. Additionally, parameters like the intermediate size and the number of layers in the FNN allow for customization of the architecture, influencing the representation of high-level features in the data. The previously mentioned hyperparameters have a parameter space that spans relatively close to the original values of the DistilBERT model, incorporating similar settings to the AutoTinyBert (Yin et al., 2021).

### 3.2. Search Method

The proposed pipeline follows a Genetic Algorithm, more specifically an NSGA-II (Deb et al., 2002) approach. The complete pipeline of the proposed search method is depicted in Figure 1. The incremental steps are outlined hereafter.

NSGA-II has been empirically shown in literature to be an effective way to apply genetic algorithms to a multi-objective problem (Lu et al., 2019). In our work modifications have been applied to the original algorithm, which were guided by intuition, empirical analysis from small-scale experiments, and constraints posed by the project's available resources. These will be explained further hereafter. Furthermore, based on empirical evidence and literature (Yin et al., 2021), the decision has been made to omit the crossover function. The crossover function would also minimize the ability of weight-inheritance (Real et al., 2017).

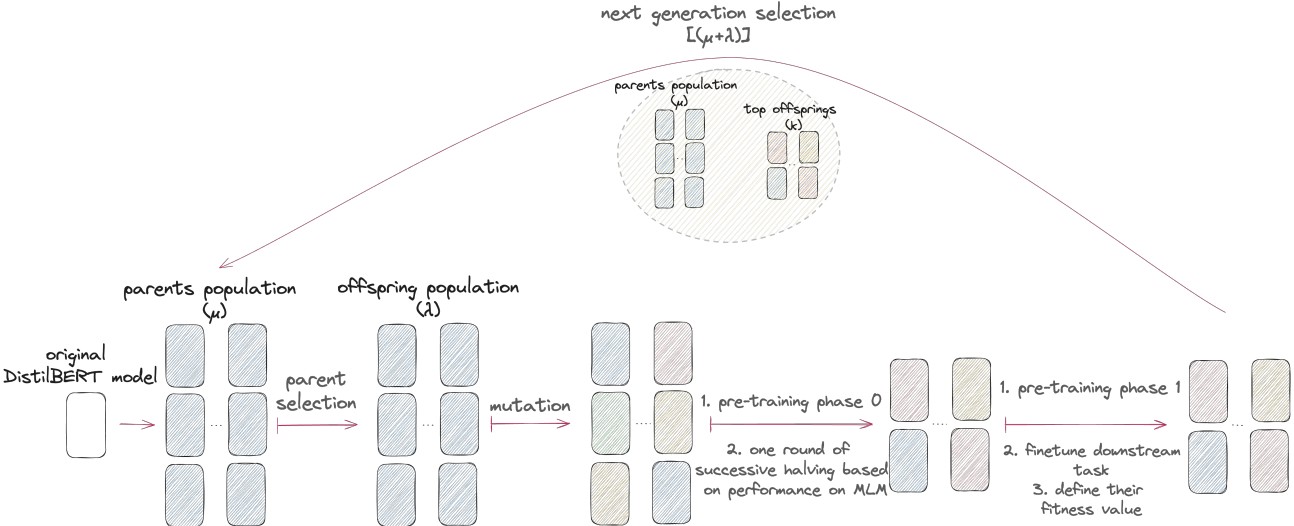

*Figure 1.* The NAS pipeline outlines the operational steps for the proposed exploration method, based on genetic algorithms. The original DistilBERT model undergoes mutations to create the initial parents' population of size $\mu$. Parent selection is then performed by investigating the Pareto front to choose the best-fitted parents. These selected parents undergo further mutations to generate the stemmed offspring population, which requires pre-training. Candidate models are pre-trained, and the top-performing half is chosen based on their masked language modeling loss. Subsequently, the selected models undergo additional pre-training and fine-tuning. Finally, we compute their objective scores and choose the next-generation parents through plus-selection.

It should be noted that due to the high cost associated with the models' life-cycle, small values were chosen for both the parents' population size ($\mu$) and the offspring population size ($\lambda$), set at $8$ and $12$, respectively.

### 3.2.1. PARENT SELECTION

Mating selection is the procedure that selects (typically two) parents that will undergo cross-over and mutation to create the offspring. Even though the mating selection procedure is typically associated with the *crossover* function, the selection of parent pairs is still implemented and will be referred to as parent selection hereafter. We implemented the parent selection procedure in order to select the best-fitted parents which will undergo mutations, to consequently create the offspring population. The parent selection procedure relies on the formulation of Pareto fronts and calculation of Pareto ranks (according to the original NSGA-II algorithm). Contrary to the original approach, the crowding distance was calculated using only one axis, promoting diversity based on the model size.

We also utilise pure tournament selection instead of the original elitist selection. This change aims to encourage exploration, especially in the absence of a *crossover* function. The $k$ value, determining the subset of the population selected for tournament selection, was set to 3. This small value enhances diversity and induces robustness in an otherwise noisy environment with small population sizes.

### 3.2.2. MUTATION

The mutation function introduces random changes to the individuals' genotype, in order to exploit identified optimal solutions. The mutation function will create our candidate models and guide the traversal through the search space with considerations about the exploration and exploitation trade-off. Firstly, we probabilistically add or remove an encoder block as a whole. Following this, the module of *multi-head self-attention* is altered stochastically in the encoder blocks, while an FNN will randomly be added or existing ones altered. Each of these modules is generated randomly within the defined parameter spaces (see Appendix A).

### 3.2.3. TRAINING PROCEDURE

As explained earlier, the mutation-operator has the potential to add or remove additional encoder blocks or adjust multi-head self-attention and FNN blocks. As a result, after a model has been mutated, it contains both already trained and untrained layers. As such, we need a process to efficiently train these models. In an attempt to make this more efficient, we split this process up in two stages. In the first phase of pre-training, *pre-training phase 0*, the unaffected layers are initially frozen and training of $4$ epochs is in effect. Through investigation of the MLM retrieved loss, we discard half of the models and proceed with the best models in the next phase, *pre-training phase 1*. We proceed to unfreeze all of the layers and further train for $8$ epochs. This

segmented training in theory should allow for more stable and efficient training, since by freezing weights the number of parameters to be updated is reduced. This makes training faster whilst also reducing the demand for computational resources. Additionally, the multi-fidelity optimization on the epoch level allows for fewer computational resources to be used per generation since we fully pre-trained fewer models.

Finally, we finetune these models on the task of question-answering using the *SQUAD v1.1* dataset. As the objective scores, we use (1) the average performance of the F1 measure (Sokolova et al., 2006) and **E**xact **M**atch (EM) scores for the model, as well as (2) the model size ratio. Since both scores (F1 and EM) are on the same scale and present equal importance in the benchmarking of this dataset, we use their average as a performance objective metric of our candidates. Depending on the application, a weighted average can be used which provides more importance on one of the two metrics. For example, in the case of the medical sector, arguably EM would be required due to the need of precise answers. Each model is now associated with performance and size objective scores, which are used in the following step, the next-generation selection.

### 3.2.4. NEXT-GENERATION SELECTION

In regards to the selection of the parents for the next generation, plus-selection, also indicated as $(\mu + \lambda)$, is used. The selection pool of candidate models for the next generation of parents can include both candidates from the offspring as well as from the parents of the current generation. This provides a form of elitism since it retains individuals potentially from the previous generation of parents, thus preventing the loss of good genetic material, ensuring that the best-performing candidates have a higher likelihood of surviving from one generation to the next. This was essential based on the small population sizes explained earlier. We select the top individuals from the pool of parents and offspring based on the Pareto rank and crowding distance (seen in Figure 2), following the elitist selection implemented in the NSGA-II approach.

## 4. Experimental Setup

We investigate the progress of the generated models over generations and derive conclusions for the effectiveness of our approach and insights into future research directions.

### 4.1. Baselines

Our baseline model is the original pre-trained DistilBERT model from HuggingFace with an EM score 62.0 and an $F1$ of 75.9, thus an average of 69.0. NAS is used to create models stemming from the original architecture, followed by the training process outlined in Section 3. To ensure a fair comparison, the same training procedures are applied to all models, and the final evaluation metrics include the candidate models downstream performance score and their number of trainable parameters (i.e. size of the model).

### 4.2. Datasets

Data required for our experimentation phase was loaded from the HuggingFace Datasets library (Lhoest et al., 2021), and these underwent necessary tokenization procedure but deliberately omitting data-level filtering or pre-processing to enhance the existing datasets. For the pre-training we used textual content extracted from articles from the English Wikipedia (Xu & Lapata, 2019) and a diverse set of books (Zhu et al., 2015), while finetuning used *SQUAD v1.1* (Rajpurkar et al., 2016). In both cases tokenization was achieved through usage the DistilBERT tokenizer, loaded from the HuggingFace Transformers library (Wolf et al., 2020).

### 4.3. Hardware

The hardware used to run the experiments is part of the SNELLIUS supercluster provided by the Dutch national ICT provider SURF, consisting of two *Intel Xeon Platinum 8360Y* CPUs, four *NVIDIA A100, 40GB HMB2* GPUs, and sixteen $32GB$ $3200MHz$ *DDR4* RAM.

### 4.4. Running time and repeats

Due to the high cost of pre-training and the limited resources, only one run of the optimisation pipeline is conducted with the number of generations limited to three. Each generation resulted in the creation of 12 offspring models, out of which 6 underwent both phases of pre-training, and the remaining 6 were cut of after 4 epochs. Additionally, the initial generation required the creation of 8 parents, which underwent both phases of pre-training. Moreover, the last generation of offspring models underwent both phases of pre-training in order to collect more data for our analysis. This resulted in the generation of 44 models in total. Based on the two phases of pre-training, 32 models have been fully trained, while 12 have only went through *pre-training phase 0*. The two pre-training phases used two GPUs with an approximate training time of 15 and 40 hours respectively. Despite the low number of generated models, insights and indications have been obtained, offering suggestions for efficient budget utilization in future research. Moreover, the pipeline has been segmented and there was independent execution of the components. This provides fine-grained control, easier expandability, or targeted future optimization and reduces the risk of invalidation of results in a resource-constrained environment.

# 5. Results

We present our results in this section to derive insights on the effectiveness of certain decision points as well as indicate effective paths for future work. The experimental analysis is presented in three distinct key areas of investigation, namely *Pareto front investigation*, *Multi-fidelity optimization* and *Cutoff criterion*. These will be expanded upon in the subsections to follow.

## 5.1. Pareto front investigation

The generation of Pareto fronts involves an iterative and repeating process. A candidate solution dominates another if it is at least as good as any other solution in the pool in all objectives and strictly better in at least one objective. The identification of non-dominated solutions categorizes them in the current Pareto front ranking. These solutions are then excluded from the set, and the process is repeated until no candidate solutions remain.

Figure 2 illustrates the plotted Pareto fronts across three generations. It should be noted that only the parents and the selected models that enter *pre-training phase 1* are depicted here, since plotted models should have underwent both phases of pre-training to retrieve the final downstream task performance. Colours indicate the Pareto front rankings for each candidate. As the Pareto front number increases (see Figure 2), the candidates represented by those Pareto fronts become less significant due to being dominated to a greater extent by candidates from lower numbered (higher ranked) Pareto fronts. Offspring candidates are represented by squares, while the parent population is denoted by circles.

While there seems to be little improvement across generations, it is evident that each generation contains a diverse set of data points, signifying the exploration of a wide range of neural network architectures by the NAS algorithm. These widely scattered data points signify a diverse set of candidate solutions, which is crucial for capturing different trade-offs between the typically conflicting objectives that we investigate. Moreover, we note that across generations the selection of models to be carried over to the next generation is typically from the parents population. It can be observed that the better performing half of the models, in each generation, always includes models of bigger size than the original. Even though models of smaller size have been successfully explored, they have not been selected to enter the second phase of pre-training (*pre-training phase 1*). This tendency, along with the lack of improvement could be attributed to the low number of explored configurations. Another attribute could be the pre-defined allocation of budget that is equal across all candidates (defined by the epochs). Absence of convergence (see Figure 3) can lead to hinderance in their selection for future generations. Gao et al. (2022) have reported exploration across more than 750

candidates, where we were able to only explore dozens of candidates within our compute budget. This further reinforces the validity of employing plus-selection, allowing us to preserve promising genetic material and preventing its loss in the event of a bad generation. The risk is quite evident due to the small population sizes.

This also emphasizes the need to explore ways that would enable bigger population pool sizes. Such a technique could be the utilization of learning curves (Mohr & van Rijn, 2022) to effectively allocate resources to more promising candidates and ensure full convergence of a smaller portion of models prior to the selection of candidates from the pool.

## 5.2. Multi-fidelity optimization

Figure 3 presents the offspring population of the third and last generation. The offspring population proceeds to undergo through both phases of pre-training (i.e. *pre-training phase 0* and *pre-training phase 1*) to derive data for the cutoff models. Three models that experienced exploding gradients or had a high valued associated loss have been excluded from the figure.

The presented learning curves of the MLM loss allow us to conclude whether the cutoff criterion between the two phases of training has been effective and whether a multi-fidelity optimization approach (such as on an epoch level) would be a good strategy to explore further. The analysis of the learning curves indicates that almost all of the selected models (solid lines) manage to remain the dominant performers after the cutoff criterion at the 4 epochs mark (around 18 000 steps). It is worth mentioning that candidate *model* 0 which was the worst performer at our cutoff point managed to outperform a significant number of models as well as two of the selected models (the two lower-performing ones). The collected data hints that learning curve analysis on LLMs would be worth investigating, potentially allowing for more models to be explored by utilizing hyperband or similar selection mechanisms.

## 5.3. Cutoff criterion

To conduct a more thorough evaluation of the effectiveness of our cutoff criterion, we investigate the correlation between the MLM loss during pre-training and the performance on the downstream task. We record the pre-training loss on the task of MLM at the cutoff point (between the two phases of pre-training) and the final pre-training loss of our offspring models in the last generation (generation 3). As previously mentioned, in order to obtain ample models for this experiment, we ensured to finish the training process for all the models of the third generation of the evolutionary algorithm for both pre-training phases.

From the 12 models in this generation, 2 had exploding

gradients. As such, we are left with 10 models. We then plot them against the performance on the downstream task of question answering. By monitoring and recording the pre-training loss both during and after training, while also examining its correlation with downstream task performance, allows us to better understand the development of model performance during the training process across a wider range of models.

Figure 4 shows the results. Note that each models is represented by two points (a red and a blue point). The points that belong to the same model can be identified by having the same value for fine-tuning performance (vertical axis). Figure 4 provides empirical indications that the loss during pre-training can be an effective cutoff criterion for model selection. The presented correlation for the final pre-training loss and the cut-off point pre-training loss in regards to the downstream task performance score is $-0.771$ and $-0.789$, respectively. These results signify strong correlation and thus the applicability of the pre-training metric score as an indication for our downstream task performance. This is crucial since during the exploration of models and their pre-training phase, we lack information on their future performance on a downstream task. An effective stopping criterion is necessary for better distribution of available resources.

## 6. Discussion

Optimization of the holistic lifecycle of LLMs through AutoML has a huge potential but it is undoubtedly accompanied by challenges (Tornede et al., 2023). Our focus is the application of NAS to language models. In this section we outline the challenges we encountered and the limitations we identified. We also propose future work.

Despite the extensive compute budget, the number of executed generations of the evolutionary algorithm is quite restricted. As evident by other black-box optimization approaches (Yin et al., 2021; Gao et al., 2022; Zoph & Le, 2017), it is typically required to generate many models until the program identifies a model that outperforms the original.

The modular pipeline provided is currently composed of two objective metrics, i.e. performance of the downstream task and the model size. However, the Pareto front optimization approach we used allows for the potential addition of further and more sophisticated objectives, such as multiple downstream tasks or memory usage. The current question-answering task is focused on the extraction of the answer, assuming the correct context is provided. Since knowledge is neither universal nor static, future applications of the pipeline or NAS should explore generative models augmented through information retrieval. Furthermore, even though empirical evidence and literature studies (Yin et al.,

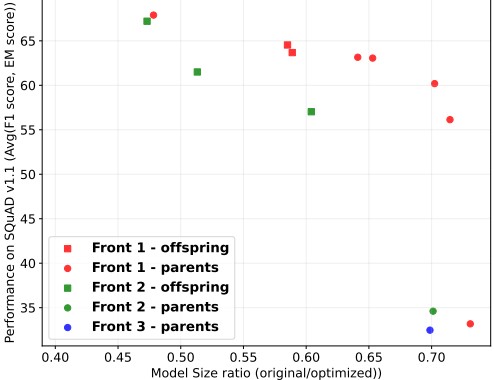

(a) Generation 1 - Pareto fronts

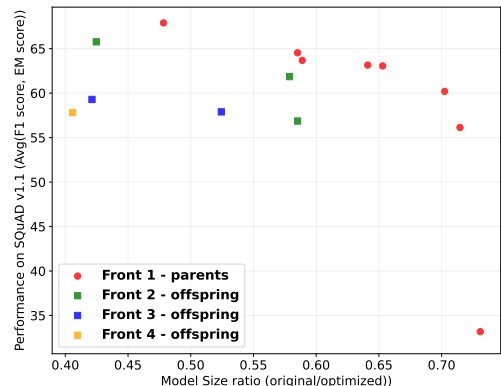

(b) Generation 2 - Pareto fronts

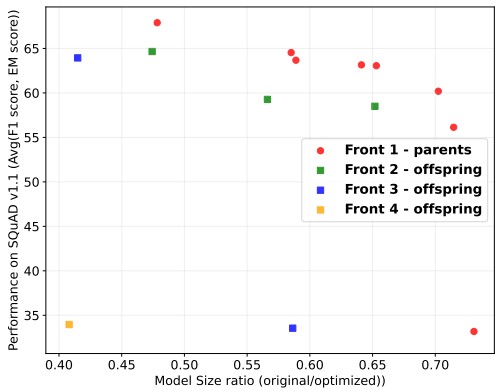

(c) Generation 3 - Pareto fronts

*Figure 2.* Pareto fronts formulated for the primary objective of the downstream task of question answering (average of F1 and Exact Match scores), and the secondary objective of the ratio of the original model size to the candidate model size. Higher numbered Pareto front rankings indicate models of less significance since they are dominated by the models in lower numbered Pareto fronts. Pareto front rankings are indicated by coloration choices specified in the legends, while population type is depicted by the shape of the data point. Circles represent candidates in the parent population, while squares depict the offspring.

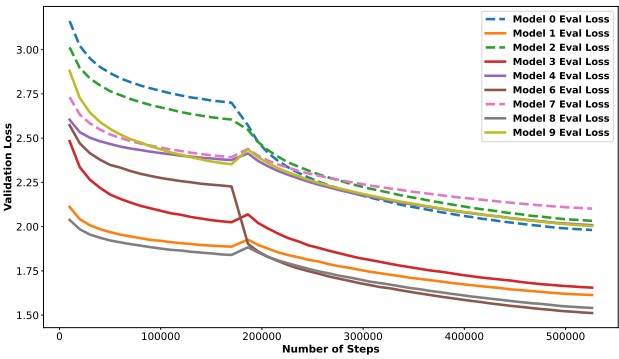

*Figure 3.* Masked language modeling task loss captured during the two phases of pre-training for the offspring population of generation 3. The solid lines represent the selected models that have moved to *pre-training phase 1*, whilst the dotted lines represent the models that were otherwise cut off.

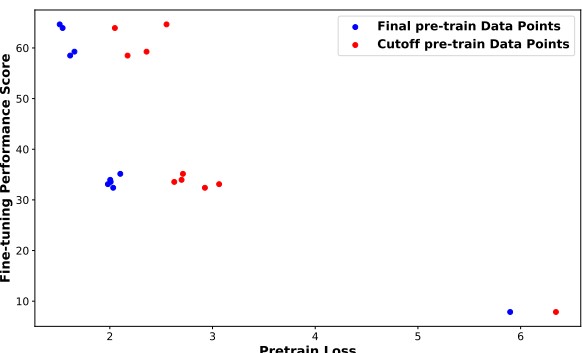

*Figure 4.* Correlation between the retrieved loss during the pre-training phase, and the downstream task performance score (post pre-training). The blue dots represent the final pre-training masked language modeling loss, and the red dots depict this loss at the cutoff point.

2021) demonstrate that the absence of a crossover function does not necessarily constrain the exploration capabilities of the black-box optimization strategy, it could be worth investigating this further.

Based on our experimental results, we believe that resource expenditure in further execution of the pipeline might yield good results, but before using these additional resources, we should explore ways to improve the training efficiency of the pipeline, in particular the pre-training component. There is currently a lot of research revolving around parameter efficient fine-tuning techniques (Ding et al., 2023) but limited intuition as to how to make the pre-training phase more efficient. It would be worth investigating techniques to leverage the retained weights of the model (of the unaffected layers) in an attempt to speed up the pre-training phase (Real et al., 2017). This can potentially allow for exploration of wider search space, utilizing the same amount of resources.

## 7. Conclusions

In this paper, we performed a case study on neural architecture search on DistilBERT. We propose a novel hierarchical search space to adapt the backbone of the transformer encoder, in combination with a segmented NSGA-II-based pipeline. Beyond reporting the final results, we also analysed the explored models. This gives insights in ways to further optimise the search procedure, and give pointers to where the pipeline can be further improved for future research. We indicated the components of the pipeline that are important to undergo further investigation and efforts should be exerted towards their optimization.

As the pre-training phase of the various models is the most expensive part of the pipeline, improving its efficiency will potentially have a large impact on future research and applications. Our results indicate that there is a correlation between the performance on the pre-training task and the downstream task with signs of applicability of learning curve utilization in a multi-fidelity optimization approach but further empirical investigation and evidence is encouraged. These insights, along with further investigation of how to efficiently use weight inheritance (Real et al., 2017) can potentially reduce the resources needed to train individuals models as well as effectively cut-off non-promising candidates. This reduction in resource expenditure can consequently boost the search space exploration.

Another interesting direction for future research can be increasing the complexity of the search space. In this work, we kept the search space relatively confined, but extending it by adding more complex mutations will result in stronger deviations from the original transformer architecture.

Additionally, subsequent research should extend our findings to include larger language models and a broader array of downstream tasks, thereby investigating and improving the generalisability of the insights.

## Acknowledgements

This research is part of the project LESSEN with project number NWA.1389.20.183 of the research program NWA-ORC 2020/21, which is (partly) funded by the Dutch Research Council (NWO). This work has been (partly) financed by the Dutch Research Council (NWO) and has used the Dutch national e-infrastructure, with the support of the SURF Cooperative, using grant no. EINF-5916. This work was partially carried out while doing a research internship at *DEUS: human(ity)-centered AI*.

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

# A. Hyperparameter search space

*Table 1.* The search space for architectural components' hyperparameters.

| Module | Probability | Action | Parameter | Parameter Space | Description |
|---|---|---|---|---|---|
| Transformer-encoder | 0.2 | Add/Remove | None | None | The default transformer-encoder block of DistilBERT is probabilistically added or removed |
| Multi-head self-attention | 0.4 | Alter | Number of attention heads | [2, 4, 8, 12, 16, 20, 24, 32] | Defines the number of attention heads that are included in the associated component, which replaces the existing one |
| FNN | 0.4 | Add/Alter | Activation function | [ReLU, GELU] | Defines the activation function to be used in between the linear layers of the FNN |
| | | | Number of layers | [2, 3, 4] | Defines the number of stacked linear layers |
| | | | Intermediate-size | [2048, 2112, 2176, ..., 3968, 4032, 4096] | Defines the dimensionality of the hidden layers of the FNN. |

