# OpenReview forum: "Resource-constrained Neural Architecture Search on Language Models: A Case Study"
_ICML.cc/2024/Workshop/WANT — WANT@ICML 2024 Poster_

### Official Review · Reviewer_mg5a · 2024-06-10
**Review: "Resource-constrained Neural Architecture Search on Language Models: A Case Study".**

**Confidence:** 4

**Summary:**

This paper investigates the application of neural architecture search to optimize the DistilBERT language model in a resource-constrained environment with a 4000 GPU-hour budget. By employing an evolutionary algorithm and a two-level hierarchical search space, the authors aim to enhance the efficiency of NAS. They segment the pre-training phase into two parts to improve resource allocation. The study finds a strong correlation between pre-training validation and downstream task performance, suggesting potential efficiency improvements in the NAS process. However, the work is limited in scale, focusing on a smaller model and evaluating a relatively small number of architectures and lacks comparison to both DistilBERT as a baseline and other established NAS methodologies.

**Strengths:**

The paper introduces a segmented pre-training phase, breaking down the common pre-training into two phases. This method offers a more efficient evaluation loop in NAS by potentially saving computational resources and improving the selection process for promising models.

Additionally, this study provides some insights into resource allocation strategies, highlighting the use of pre-training validation as a cutoff criterion and the adoption of an epoch-level stopping strategy.

**Weaknesses:**

While the architectural components defining the search space are mentioned, the paper does not provide exact values or ranges for these parameters. This omission makes the search space size unknown and hinders the reproducibility of the results.

The experiments are conducted on DistilBERT, a smaller language model, which may not generalize well to larger, more complex models. Additionally, only 44 models were evaluated during the search, which is relatively low compared to other state-of-the-art NAS studies.

The evaluation focuses solely on a single downstream task (question-answering), limiting the applicability of the findings to other NLP tasks.

The paper does not present or cite any baseline for comparison, making it unclear what the benefit of the proposed method is relative to established state-of-the-art techniques.

**Suggestions:**

Please provide detailed information on the exact values or ranges of parameters defining the search space. This will enhance the reproducibility of the results and allow for better comparison with other studies.

Extend the experiments to include other language models and a wider variety of downstream tasks to better assess the generalizability of the approach.

Present comparisons with established baselines to clearly demonstrate the advantages and improvements offered by the proposed method.

Please verify the captions in Figure 2. If the "Model Size ratio" is defined as the ratio of the original to optimized model size, and assuming the optimized model is smaller than the original, the values should logically be greater than 1.

---

### Official Review · Reviewer_Zc4S · 2024-06-12
**Very good ideas; less good methodolgy?**

**Confidence:** 4

**Summary:**

I am not an expert in LLM nor transformers but I have studied black-box optimization including NSGA-II. This paper proposes neural architecture search to find transformer architectures that maximizes masked language modelling score while minizming the number of transfromer parameters. In particular, NSGA-II is used to explore the space of feed forward layer (size, number) and multi-head attention layer of the transformer. For me the ideas and the approach proposed in this paper are great but the experimental results are not too convicing. We should accept this paper.

**Strengths:**

- Important problem
- Well written. For a non-expert in LLM/transformers; it was a super nice introduction to the field and its challenges.
- Clear and well motivated approach on the optimization objectives and the search spaces.

**Weaknesses:**

- Only 1 seed for experiments.
- Not clear to me what happens in those phase 0 and phase 1 of pre-training (page 4).

**Suggestions:**

Overall, I really like the idea of NSGA-II applied in this hierarchal way to trade-off transformer performances and sizes. But the authors decided to directly apply and study this method on a problem too big to get interesting insights. It would be amazing to try your approach on toy transformers first. In particular, since it is not clear to me if the hierarchical search space is a novel idea, using toy problems would be faster and cheaper to study.

---

### Official Review · Reviewer_G42A · 2024-06-14
**Weak Reject: The paper applies NAS to DistilBERT using NSGA-2 for evolutionary search. However, the proposed approach has limited novelty and results can be improved significantly.**

**Confidence:** 5

**Summary:**

The paper performs a case study of applying NAS to DistilBERT using NSGA-2 for evolutionary search and a hierarchical search space for the transformer architecture.

**Strengths:**

- The paper has an interesting finding that there is a good correlation between the pre-training loss and downstream task performance on SQuAD.

**Weaknesses:**

- There have been several weight-sharing NAS approaches in the literature that have used NSGA-2 for evolutionary search on Transformer based architectures. Distinctions of the proposed approach with existing methods are not clearly highlighted in the paper.
- Results were generated only on the SQuAD v1.1 dataset. Some additional results on GLUE would help strengthen the paper.
- The proposed approach seems to be compute intensive and the paper lacks comparisons with existing techniques on accuracy or compute requirements.
- Plots and figures for representing the results can be improved significantly. For example, the results in Figure-1 are confusing and difficult to understand.
- More details on the search space used are missing in the paper.
- Overall writing of the paper can be improved.

**Suggestions:**

- Generating results on certain GLUE tasks in addition to SQuAD can help strengthen the paper. Also, it might be better to include separate plots for F1 and EM scores instead of just showing the average results.
- The figures representing the results can be improved. The authors can consider including a Table that shows the trade-off between accuracy loss and model size reduction.
- More details on the search space used can be included to better understand which parameters contribute to the reduction in model size.

---

### Meta-Review · Area_Chair_WLWi · 2024-06-15

**Recommendation:** Accept (Poster)
**Confidence:** 4

**Metareview:**

The overall sentiment of the reviews appears to be borderline.

I suggest accepting it as a workshop paper if the author is willing to revise the paper based on the reviewers' feedback. The revisions should clarify the tasks (e.g., search space), training process (e.g., Figure 1), hyper-parameters, and other details. More evaluation results are expected but not mandatory.

I do hope the author can carefully think of the reviewer’s feedback, that the proposed approach indeed needs lots of computation. In this sense, it makes sense and is easier to start with smaller LLMs and toy problems.

Very much appreciate all reviewers for providing such helpful insights and suggestions.

---

### Decision · Program_Chairs · 2024-06-17

**Decision:**

Accept (Poster)

**Comment:**

We thank the authors for their time and contribution to WANT and we are pleased to share that after the reviewing process the paper has been accepted. Congratulations! We encourage the authors to consider reviewers' feedback for the improvement of the camera-ready version. We hope to see you in person at the workshop and brainstorm on efficient training research together!